# Exploring Brain and Heart Interactions during Electroconvulsive Therapy with Point-of-Care Ultrasound

**DOI:** 10.3390/medsci12020017

**Published:** 2024-03-22

**Authors:** Marvin G. Chang, Tracy A. Barbour, Edward A. Bittner

**Affiliations:** 1Department of Anesthesia, Critical Care and Pain Medicine, Massachusetts General Hospital, Harvard Medical School, Boston, MA 02114, USA; ebittner@mgb.org; 2Department of Psychiatry, Division of Behavioral Neurology and Neuropsychiatry, Massachusetts General Hospital, Harvard Medical School, Boston, MA 02114, USA; tbarbour@mgh.harvard.edu

**Keywords:** portable ultrasound, point-of-care ultrasound, POCUS, portable point-of-care ultrasound, PPOCUS, echocardiography, psychiatry, neuropsychiatry, electroconvulsive therapy, ECT, anesthesia

## Abstract

Background: Electroconvulsive therapy (ECT) is a procedure commonly used to treat a number of severe psychiatric disorders, including pharmacologic refractory depression, mania, and catatonia by purposefully inducing a generalized seizure that results in significant hemodynamic changes as a result of an initial transient parasympathetic response that is followed by a marked sympathetic response from a surge in catecholamine release. While the physiologic response of ECT on classic hemodynamic parameters such as heart rate and blood pressure has been described in the literature, real-time visualization of cardiac function using point-of-care ultrasound (POCUS) during ECT has never been reported. This study utilizes POCUS to examine cardiac function in two patients with different ages and cardiovascular risk profiles undergoing ECT. Methods: Two patients, a 74-year-old male with significant cardiovascular risks and a 23-year-old female with no significant cardiovascular risks presenting for ECT treatment, were included in this study. A portable ultrasound device was used to obtain apical four-chamber images of the heart before ECT stimulation, after seizure induction, and 2 min after seizure resolution to assess qualitative cardiac function. Two physicians with expertise in echocardiography reviewed the studies. Hemodynamic parameters, ECT settings, and seizure duration were recorded. Results: Cardiac standstill was observed in both patients during ECT stimulation. The 74-year-old patient with a significant cardiovascular risk profile exhibited a transient decline in cardiac function during ECT, while the 23-year-old patient showed no substantial worsening of cardiac function. These findings suggest that age and pre-existing cardiovascular conditions may influence the cardiac response to ECT. Other potential contributing factors to the cardiac effects of ECT include the parasympathetic and sympathetic responses, medication regimen, and seizure duration with ECT. This study also demonstrates the feasibility of using portable POCUS for real-time cardiac monitoring during ECT. Conclusion: This study reports for the first time cardiac standstill during ECT stimulation visualized using POCUS imaging. In addition, it reports on the potential differential impact of ECT on cardiac function based on patient-specific factors such as age and cardiovascular risks that may have implications for ECT and perioperative anesthetic management and optimization.

## 1. Introduction

Electroconvulsive therapy (ECT) is a procedure commonly used to treat a number of severe psychiatric disorders, including pharmacologic refractory depression, mania, and catatonia, by purposefully inducing a generalized seizure that results in significant hemodynamic changes as a result of an initial transient parasympathetic response that is followed by a marked sympathetic response from a surge in catecholamine release [1,2,3,4,5,6]. While the physiologic response of ECT on classic hemodynamic parameters such as heart rate and blood pressure has been well described in the literature, real-time visualization of cardiac function using point-of-care ultrasound (POCUS) of the heart during ECT has never been reported. Ultrasound visualization of the changes in cardiac function during ECT could provide invaluable insight that may have implications for hemodynamic optimization and anesthetic management.

Given the readily available and non-invasive nature of handheld portable POCUS and its ability to provide real-time images of the heart, it may be an ideal tool for monitoring the cardiac effects of ECT, especially in patients with pre-existing cardiovascular conditions [7,8,9,10]. Patients experience an increase in cardiac demand during ECT, and evidence suggests that cardiac complications occur most frequently in patients with underlying cardiovascular disease [2,11]. A clearer understanding of the acute changes in cardiac function during ECT could lead to improved patient safety through the optimization of anesthetic management in patients undergoing psychiatric treatment that induces significant hemodynamic and physiological effects on the brain and the heart.

In this report, we describe cardiac POCUS studies performed peri-procedurally on two patients undergoing ECT. The two patients selected were a 23-year-old female with no significant cardiovascular risk factors and a 74-year-old male with significant cardiovascular risk factors, including a history of hypertension, pulmonary embolism, deep vein thrombosis, atrial arrhythmias, and cerebrovascular accident (CVA). We hypothesized that ECT for a younger patient with no cardiovascular-associated diseases compared with an older patient with significant cardiovascular-associated risk factors would have relatively minimal effects on qualitative cardiac function, particularly left ventricular systolic function by visual inspection.

## 2. Methods

Written consent was obtained from two patients to perform cardiac POCUS during their regularly scheduled ECT treatment. The patients were counseled that the findings of this study would not provide any direct benefits to participating in this study aside from disclosure of abnormal findings observed in this study and the opportunity to visualize recordings of the cardiac images captured before, during, and after the ECT stimulus. The patients were counseled that this study would not subject them to any significant risks given that the point-of-care ultrasound imaging was non-invasive and would not distract from their clinical care and that the individual who was performing the point-of-care ultrasound imaging was separate and not a part of the anesthesia and psychiatry treatment team. No modifications were made to the usual protocol for ECT treatment.

The two patients were selected because they were at the extremes of cardiovascular risk profiles (Table 1), and the treatments were performed in succession on the same day by the same anesthesiology and psychiatry treatment team. A Butterfly IQ portable ultrasound device (Butterfly Network, Guilford, CT, USA) was used to obtain apical four-chamber images before ECT stimulation, after seizure induction, and approximately two minutes after seizure resolution [12]. A two-channel electroencephalogram (EEG) recording was performed via four electrodes placed on the forehead and mastoid processes. Two physicians (MGC and EAB) with current or prior certification in either critical care echocardiography and/or advanced echocardiography by the National Board of Echocardiography reviewed the images for qualitative assessment of cardiac function. Qualitative assessment of cardiac function was by visual assessment of the left ventricular systolic function, which has been shown to have a strong correlation with quantitative measurements of left ventricular systolic function by quantitative measurements such as the modified Simpson’s method used to measure ejection fraction [13,14]. The vital signs, which included systolic blood pressure (SBP), diastolic blood pressure (DBP), mean arterial blood pressure (MAP), oxygenation saturation (SpO_2_), and heart rate (HR) during the ECT treatment were obtained from the electronic medical record system given that these hemodynamic parameters may impact cardiac function secondary to their effects on cardiac afterload and demand, and may be an indicator of the magnitude of the autonomic response secondary to ECT. The psychiatrist who performed the ECT treatment provided data on ECT delivery parameters and seizure duration. The ECT delivery parameters included laterality of ECT electrode placement (bilateral versus unilateral) and stimulus duration, width, frequency, current, and total charge. The seizure duration was determined from the EEG data.

## 3. Results

### 3.1. Patient and ECT Characteristics

The patient characteristics, type of ECT parameters, and seizure duration are shown in Table 1.

Patient 1 was a 23-year-old female with no cardiovascular risk factors who presented for her eighth ECT procedure for the indication of depression. Patient 1’s outpatient medication regimen was atomoxetine, buspirone, clonidine, gabapentin, lamotrigine immediate release, propranolol, ramelteon, and vilazodone, all of which were taken for only psychiatric and not cardiac indications, and had been taken the day prior to ECT treatment. Right unilateral ECT with parameters shown in Table 1 resulted in a seizure duration of 33 s.

Patient 2 was a 74-year-old male with cardiovascular-associated risk factors including a history of hypertension, pulmonary embolism, deep vein thrombosis, atrial arrhythmias, and a CVA who presented for his 68th ECT treatment for the indication of depression and psychosis. Patient 2’s outpatient medication regimen was acetaminophen, amlodipine, aspirin, atorvastatin, calcium carbonate, eliquis, folic acid, magnesium, melatonin, omeprazole, sertraline, thiamine, and olanzapine, all of which had been taken the day prior to ECT treatment. Bilateral ECT was performed with parameters as shown in Table 1, resulting in a seizure duration of 85 s.

### 3.2. Real-time Cardiac Function during Seizure Induction

Video 1 shows the POCUS imaging of cardiac function for Patient 1 beginning before the ECT stimulus and continuing after stimulus termination. The POCUS imaging clip reveals a complete cardiac standstill for approximately 8 s during the stimulus. Figure 1 shows the EKG before, during, and after the ECT stimulus, demonstrating the electromagnetic interference corresponding to the cardiac standstill period. We observed similar findings in Patient 2 during the ECT stimulation period.

### 3.3. Qualitative Assessment of Cardiac Function during ECT

The cardiac POCUS video clips obtained during the ECT treatments were assessed qualitatively for cardiac function by two independent reviewers with expertise in echocardiography. Consistent with Appendix A, which illustrated POCUS imaging of cardiac function beginning prior to the ECT stimulus and continuing after stimulus termination, Patient 1 was found to have normal cardiac function prior to ECT (Appendix A), during the seizure (Appendix A), and post-seizure (Appendix A) on separate cardiac imaging. Patient 2 was found to have preserved cardiac function prior to the ECT treatment (Appendix A), significantly reduced cardiac function during the seizure (Appendix A), and normal cardiac function 2 min post-seizure (Appendix A). There was 100% consensus agreement in the qualitative assessment of cardiac function by both reviewers with echocardiography expertise.

### 3.4. Hemodynamic Response during ECT

The hemodynamic parameters, including heart rate (HR), blood pressure (BP), and oxygen saturation (SpO_2_) prior to, during, and after the seizure of each of the two patients, are shown in Table 2, Table 3 and Table 4 respectively. As typically seen, HR and BP increase immediately after the seizure and then decline to pre-seizure values. Oxygen saturation was maintained in the high 90s throughout the treatments.

## 4. Discussion

Our study reports for the first time the visual changes in cardiac function obtained with POCUS of two patients undergoing ECT with differing age and cardiovascular risk profiles. We show the real-time capture of cardiac standstill during ECT stimulation and report observations that ECT did not significantly qualitatively impact the cardiac function of a young patient without cardiovascular risks while transiently impairing cardiac function in an older patient with significant cardiovascular risks. Our study also demonstrates the feasibility of performing cardiac POCUS during ECT, which may have implications for hemodynamic optimization and anesthetic management.

### 4.1. Age and Cardiovascular Risk Factors as Determinants of Cardiac Response to ECT

Our findings underscore the possible importance of individual patient characteristics in determining the cardiac response to ECT. The older patient with multiple cardiovascular risk factors exhibited significant but transient cardiac impairment during ECT, which was not observed in the younger patient with no significant cardiovascular risk factors. Our observations suggest that age and pre-existing cardiovascular conditions may increase susceptibility to ECT-induced cardiac stress.

It is possible that the worsening in cardiac function during the seizure in Patient 2, the more elderly patient with significant cardiovascular risk factors, may have been instead secondary to an increased seizure duration and perhaps a more significant catecholamine surge compared with Patient 1, the younger patient without any cardiovascular risk factors. Rani et al. previously reported in an abstract that an increased catecholamine response was associated with an increased duration of seizures [13,15,16]. The ECT parameters, age, and gender may impact the seizure induction. Bilateral ECT has higher seizure thresholds compared with unilateral ECT, males have higher seizure thresholds compared with females, and age has varying effects on seizure threshold [17]. Thus, the bilateral ECT delivered in Patient 2 is unlikely to have been responsible for the likely more significant autonomic response observed. The outpatient medication regimens of the patients may also have played a role in limiting the cardiac response, as Patient 1 had medications that are associated with decreases in sympathetic response and seizure duration, particularly the clonidine and propranolol that she was taking for psychiatric and not cardiac indications. Of note, she had not taken these medications on the day of surgery; however, she may have had residual effects from these medications, which were taken the day prior. Patient 2 was not on any medication that would have decreased an ECT-induced sympathetic response.

It is also important to note that the much greater increase in BP and HR response during the seizure in Patient 2 may have contributed to the decline in this patient’s cardiac function secondary to increased cardiac afterload and demand, independent of the patient’s age, sex, and cardiovascular risk factors [18,19].

### 4.2. Self-Limited Nature of Impaired Cardiac Impairment during the Seizure

The impairment in cardiac function in our older patient with significant cardiovascular risks appeared to be self-limited, with his cardiac function returning to normal 2 min following the resolution of the seizure. This patient had previously received 67 ECT treatments, suggesting that the repeated ECT procedures associated with significant catecholamine surges did not result in persistent impairment in his cardiac function. The explanation for this finding may be that the catecholamine surge resulting from ECT is of limited duration, as noted by the relatively short self-limited increase in blood pressure and heart rate seen after a seizure. This short duration of increased levels of catecholamines associated with ECT treatment may contrast with more prolonged catecholamine elevations in patients who develop stress cardiomyopathy [20]. While there was no persistent effect on cardiac function in our older patient with significant cardiovascular risks, the number of treatments may play a role in the impairment of cardiac function when the heart is stressed, such as during a seizure induced by ECT.

A study of patients with normal cardiovascular function undergoing ECT by Fuenmayor et al. performed echocardiograms before and after ECT treatment and found that the ejection fraction (EF) was significantly reduced twenty minutes after ECT (average change from 63% to 52%) compared with pre-ECT treatment and that the EF returned to baseline 6 h after ECT treatment [21]. Of note, while these changes in EF observed by Fuenmayor et al. were significant quantitatively, they were not clinically significant qualitatively as the average ejection fraction, when reduced, was still considered normal [21]. It is also important to note that the study by Fuenmayor et al. did not capture real-time imaging of cardiac function during ECT treatment but rather performed echocardiograms at three sequential time points: prior to ECT treatment, 20 min after ECT treatment, and 6 h after ECT treatment [21]. Our study is novel in that it provides images of cardiac function in real time, including the transitions of seizure induction (Appendix A) through the two minutes after seizure induction and approximately two minutes after seizure resolution.

### 4.3. Feasibility of Handheld Portable POCUS in ECT

This study demonstrates the feasibility of real-time monitoring of cardiac function with POCUS during ECT. The ability to ultrasonically visualize the heart in real time during ECT provided unique insights into the changes in cardiac function that underlie the hemodynamic changes in heart rate and blood pressure that characterize ECT. It should also be noted that obtaining adequate ultrasound windows for evaluation during ECT can be technically challenging if the patient is not immobile due to the presence of myoclonus or incomplete neuromuscular blockade. Furthermore, obtaining ultrasound images is often more challenging when the patient is in the supine position, and maneuvers to optimize imaging windows, such as turning the patient in the left lateral decubitus position, could facilitate image acquisition.

The use of handheld portable point-of-care ultrasound devices as used in this study is particularly ideal for use in the ECT treatment environment because of the space limitations of compact rooms that must also accommodate an ECT machine, airway equipment, health care personnel, a patient stretcher, and the patient [7,8]. Furthermore, because of their small footprint, portable ultrasound devices are easier to clean to reduce transmission of infections [9]. While a traditional ultrasound machine is likely to have much higher resolution and provide quantitative assessment, the larger footprint of the traditional ultrasound machines would more likely impede patient care and the ability of health care providers to function effectively and efficiently [7].

### 4.4. Implications for Perioperative Anesthetic Management

The findings of this study may have significant implications for assessing, optimizing, and monitoring patients undergoing ECT treatment and anesthetic management in ECT. It is important to note that our findings are based on a small sample size and that higher-powered confirmatory studies are necessary to provide definitive guidance on perioperative anesthetic management. For high-risk patients, individualized anesthetic plans taking into account the potential for periprocedural cardiac impairment could enhance patient safety. The use of POCUS could aid in tailoring anesthesia management to mitigate the risks of hemodynamic fluctuations observed during ECT and to prevent short- and long-term complications such as cardiac ischemia, flash pulmonary edema, and cardiovascular collapse [22,23]. Furthermore, deleterious dynamic conditions, such as obstructive shock secondary to systolic anterior motion (SAM) of the mitral valve, that may be more prevalent in tachycardic and hypovolemic conditions, such as during ECT when patients have been fasting for the procedure, could be more closely monitored and optimized [24,25]. POCUS might also help identify patients who have developed stress cardiomyopathy as a result of ECT or as a result of their underlying psychiatric conditions [26]. Such information could be used to optimize patients by administering beta blockers, fluids, diuretics, vasodilators, and vasopressor support and to facilitate further diagnostic workup [27]. Furthermore, a shorter ECT stimulus may reduce the cardiac standstill time in high-risk patients. Higher-powered confirmatory studies are necessary to provide more definitive guidance on the utility of POCUS for hemodynamic optimization and identification of stress cardiomyopathy.

### 4.5. Limitations and Future Research

While this study offers valuable initial insights, it is limited by its small sample size and the qualitative nature of the cardiac assessment. Future research with larger cohorts, quantitative measures of cardiac function, and longitudinal assessments are warranted. A much larger sample of cases might be expected to enhance our understanding of cardiac risks associated with ECT, particularly in patients with varying ages and pre-existing cardiac risk factors. While the small sample size of this study limits the ability to draw more definitive conclusions, we believe that our study provides insights into the cardiac function, most interestingly, cardiac standstill observed during the ECT stimulus, which may have clinical implications in particularly high-risk patients, which should be shared with the clinical and scientific community and may lead to higher-powered confirmatory studies. Additionally, exploring the long-term cardiac effects of repeated ECT sessions in patients with varying ages and cardiovascular profiles may be valuable to determine whether some patients may require additional perioperative cardiac optimization and vigilance in care during treatment.

## 5. Conclusions

This study provides the first report of point-of-care ultrasound for real-time, continuous cardiac imaging during ECT treatment. It describes the cardiac standstill that occurred during ECT stimulation. Also, it reports on the potential differential impact of ECT on cardiac function based on patient-specific factors such as age and cardiovascular risks. These findings have implications for periprocedural care and raise new questions regarding the impact of ECT on cardiac function that warrant further investigation.

## Figures and Tables

**Figure 1 medsci-12-00017-f001:**
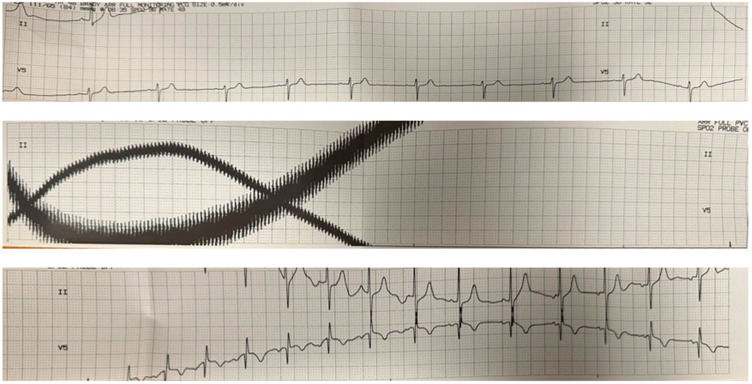
EKG during ECT procedure. The top, middle, and bottom panels show continuous EKG tracings that correspond to the times prior to the ECT stimulus, during the ECT stimulus, and after the ECT stimulus, respectively. The middle panel corresponds to the time during the ECT stimulus and shows artifacts related to electromagnetic interference from the stimulus. The significant variation in the baseline signals is likely secondary to electrical interference and motion artifact.

**Table 1 medsci-12-00017-t001:** Patient characteristics, type of ECT, and seizure duration in patients presenting for ECT treatment where cardiac POCUS was performed.

Patient	Age/Sex	Cardiovascular Risk Factors	ECT Indication/Treatment Number	ECT Parameters	Duration of Seizure
1	23 years old female	None	Depression/ 8th treatment	Right unilateral, stimulus duration 8 s, pulse width 0.4 ms, frequency 110 Hz, current 800 mA, total charge delivered 563.2 mC	33 s
2	74 years old male	hypertension, pulmonary embolism, atrial arrhythmias, and cerebrovascular accident (CVA)	Depression and psychosis/ 68th treatment	Bilateral, stimulus duration 6 s, pulse width 0.5 ms, frequency 40 Hz, current 800 mA, total charge delivered 192 mC	85 s

**Table 2 medsci-12-00017-t002:** Heart rate (HR) response before, during, and after the seizure.

Patient	Pre-Seizure HR	Seizure HR	Post-Seizure HR
1	85	111	96
2	79	118	67

**Table 3 medsci-12-00017-t003:** Blood pressure (BP) response before, during, and after the seizure. BP is exhibited in SBP/DBP (MAP), where SBP is systolic blood pressure, DBP is diastolic blood pressure, and MAP is mean arterial pressure.

Patient	Pre-Seizure BP	Seizure SBP	Post-Seizure BP
1	107/71 (83)	140/90 (106)	98/53 (71)
2	152/74 (106)	196/107 (141)	151/70 (102)

**Table 4 medsci-12-00017-t004:** Oxygen saturation (SpO2) before, during, and after the seizure.

Patient	Pre-Seizure SpO_2_	Seizure SpO_2_	Post-Seizure SpO_2_
1	97	100	96
2	99	99	96

## Data Availability

The original contributions presented in the study are included in the article/Appendix A, further inquiries can be directed to the corresponding authors.

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
