# Peer review of "Exploring Brain and Heart Interactions during Electroconvulsive Therapy with Point-of-Care Ultrasound"

_medsci, 2024, doi:10.3390/medsci12020017_

Round 1
Reviewer 1 Report
Comments and Suggestions for Authors
This manuscript addresses the clinically important question of the effects of ECT on cardiac function, using Point of Care Ultrasound (POCUS). Use of POCUS in a clinical ECT setting is feasible. The time scale of the procedure that allows observation immediately before, during and after the seizure.
The authors report data in two cases: a young person with no cardiac risks and an elderly person with multiple risk factors. Of interest, cardiac standstill was observed during the seizure in both cases. In accord with expectation the young patient showed no evidence of continuing cardiac dysfunction, whereas the elderly patient exhibited cardiac dysfunction during seizure and immediately afterwards, though cardiac function in this patient returned to normal after 2 seconds. It is noteworthy the elderly patient exhibited very high blood pressure (196/107) during the seizure.
The major value of this study is the demonstration that POCUS is feasible and provides clinically informative information, though in the two cases reported in this paper, the findings would not be likely to have influenced clinical management.
Nonetheless, in light of the reluctance of clinicians to administer ECT in cases with substantial cardiac risk factors, the feasibility of POCUS offers a potentially useful tool for monitoring cardiac function, and might in principle guide adjustment of ECT delivery during a course of treatment.
The fact that only two cases were reported in this study is a limitation acknowledged by the authors. Study of a much larger sample of cases might be expected to provide a useful enhancement of our understanding of cardiac risks associated dwt ECT, especially in cases with known pre-exiting cardiac risk factors.
Author Response
Response: Thank you so much for taking the time to review our manuscript and providing constructive feedback. We agree with the reviewer’s statement that a much larger sample of cases might be expected to provide a useful enhancement of our understanding of cardiac risks associated with ECT, particularly in patients with preexisting cardiac risk factors. We have included this helpful point provided by the reviewer in our limitations section, and have added the following statements: “Future research with larger cohorts, quantitative measures of cardiac function, and longitudinal assessments are warranted. A much larger sample of cases might be expected to enhance our understanding of cardiac risks associated with ECT, particularly in patients with varying ages and preexisting cardiac risk factors. While the small sample size of this study limits the ability to draw more definitive conclusions, we believe that our study provides insights into the cardiac function, most interestingly, cardiac standstill observed during the ECT stimulus, which may have clinical implications in particularly high-risk patients, which should be shared with the clinical and scientific community and may lead to higher powered confirmatory studies.”
Reviewer 2 Report
Comments and Suggestions for Authors
The manuscript is well written. Comments and suggestions:
Introduction
- The introduction provides a lot of detailed background on the physiologic effects of ECT, but the actual purpose and focus of the study is not clearly stated upfront. The authors do not introduce the use of POCUS for monitoring cardiac function during ECT until later in the introduction. The objective should be stated more clearly early on.
- The writing could be more concise in places - some sentences are long and a bit convoluted. Simplifying wording could improve clarity.
- When introducing the two patient cases, more clinical details would provide helpful context, especially for the 74-year-old male with cardiovascular risk factors. What specifically were those risk factors?
- The hypothesis compares effects on "qualitative cardiac function" between the two patients, but qualitative function is vague. The authors should define more specifically what cardiac parameters they are assessing.
- Using only two patient cases is an extremely small sample size with limited ability to draw conclusions or test hypotheses related to age and cardiovascular status. Expanding to more patients in each age/risk category would strengthen the analysis.
- There is no mention of the specific POCUS measurements obtained or how cardiac function was quantitatively or qualitatively assessed using the imaging data. This methodology needs to be clearly described.
Methods
- More details should be provided on the consent process and what was communicated to patients about risks/benefits of participating. How was patient privacy protected?
- The methods describe obtaining images at 3 time points but do not define what parameters of cardiac function were evaluated qualitatively at each point. Specific metrics should be stated.
- Using a single physician to review images could introduce bias. Having 2 independent reviewers and assessing inter-rater reliability would strengthen methodology.
- No details provided on vital signs - what specifically was measured and what constituted hemodynamic changes of interest? How was EEG data used?
- The writing lacks clarity - "per usual care" and "data on ECT delivery parameters" are vague. More specifics needed on procedures and metrics.
- No information provided on key factors that could impact cardiac function like medications patients were taking. Were any modifications to usual protocols made?
Discussion and Conclusion
- The sample size of two patients is extremely small, making it very difficult to draw any definitive conclusions about the role of age and cardiovascular risk factors in determining cardiac response to ECT. Many more patients would need to be studied.
- The assessment of cardiac function is stated to be qualitative, but details are not provided on what specific aspects of function were evaluated. Quantitative measurements of function (e.g. ejection fraction) would strengthen the analysis.
- Confounding factors that could influence cardiac response, beyond just age and CVD risk, are not discussed. For example, differences in ECT stimulation parameters, seizure duration, medications, etc. between the two patients.
- The proposed implications for anesthetic management are speculative given the very limited data. Recommendations about individualized anesthesia plans are premature without more robust evidence on ECT's cardiovascular effects.
- The feasibility of using POCUS during ECT is demonstrated in two patients only. More evidence is needed to support claims about its utility for hemodynamic optimization and identifying stress cardiomyopathy before drawing strong conclusions.
- Writing could be tighter in places - there is repetition across sections, and some lengthy, convoluted sentences.
In summary, this initial study demonstrates the potential for using POCUS to study cardiac effects of ECT, but substantially more investigation is required to determine definitive links between patient factors, cardiac function changes, and clinical decision making around ECT procedures. The conclusions reached based on two subjects are vastly overstated.
Author Response
Thank you so much for taking the time to review our manuscript and providing constructive feedback. Please find below our responses to your helpful comments and suggestions.
The manuscript is well written. Comments and suggestions:
Introduction
- The introduction provides a lot of detailed background on the physiologic effects of ECT, but the actual purpose and focus of the study is not clearly stated upfront. The authors do not introduce the use of POCUS for monitoring cardiac function during ECT until later in the introduction. The objective should be stated more clearly early on.
Response: Thank you for raising this important point. We have shortened the initial paragraph of the introduction to more clearly state the objective earlier on. The first paragraph of the introduction has been revised to the following: “Electroconvulsive therapy (ECT) is a procedure commonly used to treat a number of severe psychiatric disorders, including pharmacologic refractory depression, mania, and catatonia, by purposefully inducing a generalized seizure that results in significant hemodynamic changes as a result of an initial transient parasympathetic response that is followed by a marked sympathetic response from a surge in catecholamine release. While the physiologic response of ECT on classic hemodynamic parameters such as heart rate and blood pressure has been well described in the literature, real-time visualization of cardiac function using point of care ultrasound (POCUS) of the heart during ECT has never been reported. Ultrasound visualization of the changes in cardiac function during ECT could provide invaluable insight that may have implications for hemodynamic optimization and anesthetic management.
- The writing could be more concise in places - some sentences are long and a bit convoluted. Simplifying wording could improve clarity.
Response: Thank you for the very helpful feedback. We agree that the writing could be more concise in place and the some sentences are long and convoluted. We have made significant efforts to simplify our wording and improve clarity. Of note, we have also used the Grammarly.com service to improve grammar and clarity of our manuscript. - When introducing the two patient cases, more clinical details would provide helpful context, especially for the 74-year-old male with cardiovascular risk factors. What specifically were those risk factors?
Response: Thank you for providing this valuable recommendation to improve the context in the introduction. We have now included these cardiovascular associated risk factors in the introduction. The following statement has been modified as follows: “The two patients selected were a 23-year-old female with no significant cardiovascular risk factors and a 74-year-old male with significant cardiovascular risk factors, including a history of hypertension, pulmonary embolism, deep vein thrombosis, atrial arrhythmias, and cerebrovascular accident (CVA).” - The hypothesis compares effects on "qualitative cardiac function" between the two patients, but qualitative function is vague. The authors should define more specifically what cardiac parameters they are assessing.
Response: Thank you for raising this important and valid point. We assessed qualitative cardiac function, in particular left ventricular systolic function by visual assessment given that the portable ultrasound machine used in the study is unable to make quantitative measurements and relies on qualitative assessment of left ventricular systolic function. Of note, left ventricular systolic function by visual estimation has been shown to strong correlation with quantitative measurements of left ventricular systolic function by quantitative measurements such as the modified Simpson’s method used to measure ejection fraction. We have clarified this important point raised by the reviewer in the introduction with the following modified statement: “We hypothesized that ECT for a younger patient with no cardiovascular-associated diseases compared to an older patient with significant cardiovascular-associated risk factors would have relatively minimal effects on qualitative cardiac function, particularly left ventricular systolic function by visual inspection.” We have also added the following statement to the methods section: “Qualitative assessment of cardiac function was by visual assessment of the left ventricular systolic function, which has been shown to have a strong correlation with quantitative measurements of left ventricular systolic function by quantitative measurements such as the modified Simpson’s method used to measure ejection fraction.” - Using only two patient cases is an extremely small sample size with limited ability to draw conclusions or test hypotheses related to age and cardiovascular status. Expanding to more patients in each age/risk category would strengthen the analysis.
Response: We agree with the reviewer on this important point. While this point is true, we believe that it does provide insights into cardiac function, most interestingly cardiac standstill observed during the ECT stimulus which may have clinical implications in particularly high risk patients and believe that this finding should be shared with the clinical and scientific community. We agree that that our sample size is not large enough to make more definitive conclusions from our findings, however, believe that our study may lead to higher powered confirmatory studies. We have acknowledged these points in our limitations and have added the following statement: “While the small sample size of this study limits the ability to draw more definitive conclusions, we believe that our study provides insights into the cardiac function, most interestingly, cardiac standstill observed during the ECT stimulus, which may have clinical implications in particularly high-risk patients, which should be shared with the clinical and scientific community and may lead to higher powered confirmatory studies.” - There is no mention of the specific POCUS measurements obtained or how cardiac function was quantitatively or qualitatively assessed using the imaging data. This methodology needs to be clearly described.
Response: Thank you for raising this important and valid point. We have addressed the above in point #4 in the introduction section by the reviewer.
Methods
- More details should be provided on the consent process and what was communicated to patients about risks/benefits of participating. How was patient privacy protected?
Response: Thank you for inquiring about this information. The patients were counseled that the study findings would not provide any direct benefits to participating in the study aside from disclosure of abnormal findings observed in the study and the opportunity to visualize recordings of the cardiac images captured prior to, during and after the ECT stimulus. The patients were counseled that the study would not subject them to any significant risks given that the point of care ultrasound imaging was non-invasive and would not distract from their clinical care, given that the individual who was performing the point of care ultrasound imaging was separate and not a part of the anesthesia and psychiatry treatment team. Patient privacy was protected by providing the minimum necessary amount of information for this report. We have included these additional points in the methods section with the following statements: “The patients were counseled that the study findings would not provide any direct benefits to participating in the study aside from disclosure of abnormal findings observed in the study and the opportunity to visualize recordings of the cardiac images captured before, during, and after the ECT stimulus. The patients were counseled that the study would not subject them to any significant risks given that the point of care ultrasound imaging was non-invasive and would not distract from their clinical care, given that the individual who was performing the point of care ultrasound imaging was separate and not a part of the anesthesia and psychiatry treatment team. No modifications were made to the usual protocol for ECT treatment.” - The methods describe obtaining images at 3 time points but do not define what parameters of cardiac function were evaluated qualitatively at each point. Specific metrics should be stated.
Response: Thank you for raising this important and valid point. We have addressed this in the reviewer’s point #4 of the introduction section by the reviewer above. - Using a single physician to review images could introduce bias. Having 2 independent reviewers and assessing inter-rater reliability would strengthen methodology.
Response: Thank you for raising this important point. Two physicians independently reviewed the images with current or prior certification in critical care echocardiography and/or advanced echocardiography by the National Board of Echocardiography. We have included the following statement in the methods section: Two physicians (MGC and EAB) with current or prior certification in either critical care echocardiography and/or advanced echocardiography by the National Board of Echocardiography reviewed the images for qualitative assessment of cardiac function.” In the results section, we have also added the following statement: “There was 100% consensus agreement in the qualitative assessment of cardiac function by both reviewers with echocardiography expertise.” - No details provided on vital signs - what specifically was measured and what constituted hemodynamic changes of interest? How was EEG data used?
Response: Thank you for the helpful clarifications regarding vital signs and use of EEG data. The vital signs that were measured and obtained from the electronic medical record system included systolic blood pressure, diastolic blood pressure, mean arteria blood, pressure, oxygen saturation, and heart rate. Hemodynamic changes related to increases in heart rate and blood pressure and decreases in oxygen saturation following ECT stimulus may impact cardiac function in particular given that these hemodynamic parameters may adversely impact cardiac function secondary to increase cardiac afterload and demand, and may be an indirect measurement of the magnitude of the autonomic response secondary to ECT. We have modified the statement in the methods as follows: “The vital signs, which included systolic blood pressure (SBP), diastolic blood pressure (DBP), mean arterial blood pressure (MAP), oxygenation saturation (SpO2), and heart rate (HR) during the ECT treatment were obtained from the electronic medical record system given that these hemodynamic parameters may impact cardiac function secondary to their effects on cardiac afterload and demand, and may be an indicator of the magnitude of the autonomic response secondary to ECT.” We have modified the statement in the discussion as follows: “It is also important to note that the much greater increase in BP and HR response during the seizure in Patient 2 may have contributed to the decline in this patient’s cardiac function secondary to increased cardiac afterload and demand, independent of the patient’s age, sex, and cardiovascular risk factors.” The EEG data was used to determine the seizure activity and duration, and we have added the following statement in the methods to clarify this: “The seizure duration was determined from the EEG data.” - The writing lacks clarity - "per usual care" and "data on ECT delivery parameters" are vague. More specifics needed on procedures and metrics.
Response: Thank you for the useful feedback. We have removed the “per usual care” from the statement in the methods which now reads as follows: “A two channel electroencephalogram (EEG) recording was performed via four electrodes placed on the forehead and mastoid processes.” We have provided more details on the ECT delivery parameters as well. The following statement in the methods has been added: “The ECT delivery parameters included laterality of ECT electrode placement (bilateral versus unilateral) and stimulus duration, width, frequency, current, and total charge.” - No information provided on key factors that could impact cardiac function like medications patients were taking. Were any modifications to usual protocols made?
Response: Thank you for raising this important point. We agree with the review that it is important to provide key factors that could impact cardiac function like medications. We have provided these details in the manuscript and have included the following statements: “. Patient 1’s outpatient medication regimen was atomoxetine, buspirone, clonidine, gabapentin, lamotrigine immediate release, propranolol, ramelteon, and vilazodone, all of which were taken for only psychiatric and not cardiac indications and had been taken the day prior to ECT treatment.” And “Patient 2 outpatient medication regimen was acetaminophen, amlodipine, aspirin, atorvastatin, calcium carbonate, eliquis, folic acid, magnesium, melatonin, omeprazole, sertraline, thiamine, and olanzapine, all of which had been taken the day prior to ECT treatment.” No modifications were made to the usual protocols, and have provided a statement on this in the methods as follows: “No modifications were made to the usual protocol for ECT treatment.”
Discussion and Conclusion
- The sample size of two patients is extremely small, making it very difficult to draw any definitive conclusions about the role of age and cardiovascular risk factors in determining cardiac response to ECT. Many more patients would need to be studied.
Response: Thank you for raising this important and valid point. We have addressed this in the reviewer’s point #5 of the introduction section by the reviewer above. - The assessment of cardiac function is stated to be qualitative, but details are not provided on what specific aspects of function were evaluated. Quantitative measurements of function (e.g. ejection fraction) would strengthen the analysis.
Response: Thank you for raising this important and valid point. We have addressed this in the reviewer’s point #4 of the introduction section by the reviewer above.
- Confounding factors that could influence cardiac response, beyond just age and CVD risk, are not discussed. For example, differences in ECT stimulation parameters, seizure duration, medications, etc. between the two patients.
Response: Thank you for raising this important point. We agree with the reviewer that other confounding factors other than age and CVD risk could influence cardiac response. We have included the following statements in the discussion to address this: “It is possible that the worsening in cardiac function during the seizure in Patient 2, the more elderly patient with significant cardiovascular risk factors, may have been instead secondary to an increased seizure duration and perhaps a more significant catecholamine surge compared to Patient 1, the younger patient without any cardiovascular risk factors. Rani et al. previously reported in an abstract that an increased catecholamine response was associated with an increased duration of seizures. The ECT parameters, age, and gender may impact the seizure induction. Bilateral ECT has higher seizure thresholds compared to unilateral ECT, men have higher seizure thresholds compared to females, and age has varying effects on seizure threshold. Thus, the bilateral ECT delivered in Patient 2 is unlikely to have been responsible for the likely more significant autonomic response observed. The outpatient medication regimens of the patients may also have played a role in limiting the cardiac response, as Patient 1 had medications that are associated with decreases in sympathetic response and seizure duration, particularly the clonidine and propranolol that she was taking for psychiatric and not cardiac indications. Of note, she had not taken these medications on the day of surgery; however, she may have had residual effects from these medications taken the day prior. Patient 2 was not on any medication that would have decreased an ECT-induced sympathetic response.
- The proposed implications for anesthetic management are speculative given the very limited data. Recommendations about individualized anesthesia plans are premature without more robust evidence on ECT's cardiovascular effects.
Response: Thank you for raising this important point. We agree with the reviewer that recommendations about individualized anesthesia plans are premature without more robust evidence on ECT’s cardiovascular effects. We have added/modified the statements in the section 4.4 on implications for perioperative anesthetic management as follows: “The study findings may have significant implications for assessing, optimizing, and monitoring patients undergoing ECT treatment and anesthetic management in ECT. It is important to note that our findings are based on small sample size and that higher-powered confirmatory studies are necessary to provide definitive guidance on perioperative anesthetic management.”
- The feasibility of using POCUS during ECT is demonstrated in two patients only. More evidence is needed to support claims about its utility for hemodynamic optimization and identifying stress cardiomyopathy before drawing strong conclusions.
Response: Thank you for raising this important point. We agree with the reviewer that more evidence is needed to support claims about its utility for hemodynamic optimization and identifying stress cardiomyopathy before drawing strong conclusions, and have added the following statement: “Higher-powered confirmatory studies are necessary to provide more definitive guidance on the utility of POCUS for hemodynamic optimization and identifying stress cardiomyopathy.” - Writing could be tighter in places - there is repetition across sections, and some lengthy, convoluted sentences.
Response: Thank you for raising this point. We agree with the reviewer on this point and have made efforts to reduce the repetitions across sections and limit length and convoluted sentences. Of note, we have also used the Grammarly.com service to improve grammar and clarity of our manuscript.
Reviewer 3 Report
Comments and Suggestions for Authors
The purpose of the study was to examine cardiac function using point of care ultrasound (POCUS) of the heart during electroconvulsive therapy (ECT). To accomplish this a dyad case study involving 2 participants was utilized. Importantly, the 2 participants were of greatly different ages and cardiovascular risk factors. Specifically, a 74 year old male with significant cardiovascular risks and a 23 year old female with no significant cardiovascular risks were recruited during their normal ECT sessions they were already doing. Thus, the study was novel as cardiovascular function has not been measured with POCUS during ECT before to my knowledge. Only basic measures of heart rate, blood pressure, etc have been done as mentioned by the authors.
The authors predicted that the ECT for the young patient would have minimal effects on cardiovascular function whereas the older patient would experience significantly greater effects on qualitative cardiac function. Overall, the results were essentially in line with these hypotheses as various measures of cardiac function were impaired in the older subject, albeit for a relatively short time. The younger subject did not experience major changes in cardiac function during ECT.
Overall, the study seemed to be conducted carefully, was easy to understand, had a solid design, and was very well-written with few grammatical or typographical errors. I think the study adds to the literature on the topics of ECT, brain stimulation, and cardiac function. Limitations were acknowledged. The study should be of interest to readers of the journal and researchers is several related fields. The results will likely lead to further appropriately-powered work in the future on the topic.
Overall, I don’t think the study had any fatal flaws and the only major weakness was the fact it was a case study involving 2 subjects. I only have a few minor comments of various types that the authors should consider and a few minor corrections to suggest.
- 4th line and throughout the paper. There needs to be a space before the bracket of all the references as there is no space before it and the previous word on all the citations in the text.
- I know the author touched very briefly in the methods and the discussion of these topics, but it may be better to add a little bit more of why these 2 participants were selected, why the differences in bilateral vs unilateral ECT, why this may or may not matter etc. Same thing for the duration of the seizure. I think more information on how this may have affected results is needed.
- Bibliography: It appears that there are formatting inconsistencies in the bibliography. Some the first letter of all words of journal article titles are capitalized in some references and other times all words except the first one are not. There may also be inconsistencies in how the journal titles are formatted. Finally, the number of references is low even for a case study.
minor proofreading/formatting
Author Response
Thank you so much for taking the time to review our manuscript and providing constructive feedback. Please find below our responses to your helpful comments and suggestions.
The purpose of the study was to examine cardiac function using point of care ultrasound (POCUS) of the heart during electroconvulsive therapy (ECT). To accomplish this a dyad case study involving 2 participants was utilized. Importantly, the 2 participants were of greatly different ages and cardiovascular risk factors. Specifically, a 74 year old male with significant cardiovascular risks and a 23 year old female with no significant cardiovascular risks were recruited during their normal ECT sessions they were already doing. Thus, the study was novel as cardiovascular function has not been measured with POCUS during ECT before to my knowledge. Only basic measures of heart rate, blood pressure, etc have been done as mentioned by the authors.
The authors predicted that the ECT for the young patient would have minimal effects on cardiovascular function whereas the older patient would experience significantly greater effects on qualitative cardiac function. Overall, the results were essentially in line with these hypotheses as various measures of cardiac function were impaired in the older subject, albeit for a relatively short time. The younger subject did not experience major changes in cardiac function during ECT.
Overall, the study seemed to be conducted carefully, was easy to understand, had a solid design, and was very well-written with few grammatical or typographical errors. I think the study adds to the literature on the topics of ECT, brain stimulation, and cardiac function. Limitations were acknowledged. The study should be of interest to readers of the journal and researchers is several related fields. The results will likely lead to further appropriately-powered work in the future on the topic.
Overall, I don’t think the study had any fatal flaws and the only major weakness was the fact it was a case study involving 2 subjects. I only have a few minor comments of various types that the authors should consider and a few minor corrections to suggest.
- 4th line and throughout the paper. There needs to be a space before the bracket of all the references as there is no space before it and the previous word on all the citations in the text.
Response: Thank you for raising this important issue with the citations. We have placed a space before the bracket of all references. - I know the author touched very briefly in the methods and the discussion of these topics, but it may be better to add a little bit more of why these 2 participants were selected, why the differences in bilateral vs unilateral ECT, why this may or may not matter etc. Same thing for the duration of the seizure. I think more information on how this may have affected results is needed.
Response: Thank you for raising these important points. We have modified the manuscript to include more details on why these two participants were selected. In the methods section, we added the following statement “The two patients were selected because they were at the extremes of cardiovascular risk profiles (Table 1), and the treatments were performed in succession on the same day by the same anesthesiology and psychiatry treatment team.” We have also added more details about whether differences in seizure duration and ECT delivery via bilateral versus unilateral electrode placement may have mattered. We have added the following to the discussion: “Rani et al. previously reported in an abstract that an increased catecholamine response was associated with an increased duration of seizures. The ECT parameters, age, and gender may impact the seizure induction. Bilateral ECT has higher seizure thresholds compared to unilateral ECT, men have higher seizure thresholds compared to females, and age has varying effects on seizure threshold. Thus, the bilateral ECT delivered in Patient 2 is unlikely to have been responsible for the likely more significant autonomic response observed.”
Bibliography: It appears that there are formatting inconsistencies in the bibliography. Some the first letter of all words of journal article titles are capitalized in some references and other times all words except the first one are not. There may also be inconsistencies in how the journal titles are formatted. Finally, the number of references is low even for a case study.
Response: Thank you for raising this issue. We had originally imported these references using endnote by selecting reference by PMIDs. We have now manually modified the titles so that they are consistent.
Reviewer 4 Report
Comments and Suggestions for Authors
Very interesting findings, illustrative videos. Acceptable after minor revision, ie, after reordering the presentation of the videos: In section 3.3, please describe the contents of video 1 before that of video 2 etc.
Author Response
Response: Thank you very much for taking the time to review our manuscript and provide constructive feedback. We thank the review for raising the issue with the videos. We have referenced Video 1 in Section 3.3 prior to discussing videos 2-7. The following statement has been modified in Section 3.3: “Consistent with Video 1, which illustrated POCUS imaging of cardiac function beginning prior to the ECT stimulus and continuing after stimulus termination, Patient 1 was found to have normal cardiac function prior to ECT (Video 2), during the seizure (Video 3) and post-seizure (Video 4) on separate cardiac imaging.” Of note, Video 1 is also reference in the prior section 3.2 as well. Thank you for recommending these valuable improvements to our manuscript.
Round 2
Reviewer 2 Report
Comments and Suggestions for Authors
The authors responded to my comments very well. Thank you.